# Healthcare Professionals’ and Users’ Experiences of Intersectoral Care between Hospital and Community Mental Healthcare

**DOI:** 10.3390/ijerph17186510

**Published:** 2020-09-07

**Authors:** Kim Jørgensen, Mette Bonde Dahl, Jesper Frederiksen

**Affiliations:** 1The Research Collaboration, Psychiatric Centre North Zealand, Dyrehavevej 48, 3400 Hillerød, Denmark; 2Department of Occupational Therapy and Nursing, University College Lillebælt–Business Academy and Professional College, Niels Bohrs Allé 1, 5230 Odense, Denmark; mbda@ucl.dk; 3Centre for Nursing, University College Absalon, Trekroner Forskerpark 4, 4000 Roskilde, Denmark; jefr@pha.dk

**Keywords:** community mental healthcare, hospitals’ mental healthcare, healthcare professionals, intersectoral care, nurse, multiprofessional care, patient participation, recovery, users

## Abstract

This paper explores healthcare professionals’ and users’ experience of coherent intersectoral care between hospital mental healthcare and community mental healthcare. A total of 20 healthcare professionals, primarily nurses, and 14 users with a range of mental illnesses participated in nine focus group interviews (FGIs). Participants were encouraged in the FGIs to reflect upon their experience of coherency in intersectoral care. The analysis of FGIs was informed by a phenomenological-hermeneutic approach in a research group from 2016–2019. The Consolidated Criteria for Reporting Qualitative Research checklist was used as a guideline to ensure complete and accurate reporting of the study. The analysis led to the generation of several themes from a professional perspective and from a user perspective, addressed barriers to coherent intersectoral care. The healthcare professionals experienced barriers such as a lack of common language and knowledge of partners. The users did not feel involved and lacked coherence in their recovery processes and, as such, intersectoral care was often experienced as being lost in a maze.

## 1. Introduction

In Western countries, coherent intersectoral care is one of the cornerstones of the healthcare debate. It requires a focus on long-term care instead of concentrating only on the here and now in one sector [1]. According to healthcare policy, coherent intersectoral care is presented as a stated objective, with requirements for healthcare professionals to ensure that users systematically experience coherent efforts across the long-term process through hospital mental healthcare and community mental healthcare [2]

At the political level, a good coherent intersectoral care service through improving collaboration is seen as key to improving and developing the mental healthcare system [3,4]. However, despite such political aims, great awareness, and many initiatives, there are still challenges within and criticisms of the mental healthcare system in Scandinavia [5] and other Western countries. This can, for instance, be explained as a lack of coherence in the users’ experience of the cross-sectoral process [6,7,8].

The Ministry of Health and the Elderly defines the measure of coherent intersectoral care as the sum of the activities, contacts, and events in the health service that a user or a defined group of users experiences in relation to the healthcare service [9]. In some studies, it is assumed that ‘intersectoral collaboration’ is an important tool in achieving coherence. Additionally, coherent intersectoral care must be based on a recovery-oriented approach [10,11]. There are only a few studies that focus on coherent intersectoral care in a psychiatric context [12,13,14,15,16,17,18]. These studies highlight issues concerning inadequate communication and coordination across the professional services [19]. A number of studies show a lack responsibility for the users’ continuous long-term process between the two sectors, and the efforts are not experienced as coherent from a user’s perspective [18,20]. Important information about the user’s situation, plans, etc., is lost in the process, e.g., from discharge from hospital mental healthcare to community mental healthcare [10]. The existing research is defined by primarily focusing on patient treatment within one sector or focusing only on the actual transition between the sectors [18]. A lack of coherent intersectoral care is not just problematic in mental healthcare, the same challenges occur for people with chronic illnesses, but in mental healthcare, this is particularly relevant because vulnerable users and psychiatry are under-prioritized. Our assumption is therefore that the challenges of coherence of intersectoral care may appear as manifest and relevant to the entire health sector.

Some studies highlight problem areas such as insufficient information-sharing across sectors, uncertainty about who is responsible for healthcare before, during, and after discharge, as well as other coordination and cooperation issues [21]. Users often experience having to tell their story over and over again; they do not feel that the healthcare professionals are treating them as whole persons [22,23].

Another challenge is that users are repeatedly discharged from hospital mental healthcare to community mental healthcare, which places greater responsibility upon healthcare professionals in community mental healthcare [24,25]. Reports from user associations show that, as a result of the lack of coherency throughout the intersectoral care, some users experience being haphazardly transferred from one set of healthcare professionals to another [26].

There is some grey literature on the subject, including a report showing that users of psychiatric services experience that they are in contact with many different professionals within regions and municipalities, which creates a great deal of complexity. It is described as a great challenge to find your way around the system, partly because professional divisions and sector divisions do not immediately make sense in a perspective that concerns making everyday life work and relates to shaping the best possible life with a long-term mental illness in the background. Coherence for citizens is thus not simply linked to continuity and smooth transitions between sectors and professionals.

The challenges caused by divided administration, economics, and differing legislation, values, and norms all affect and test coordinated efforts to improve coherency for users [25,27,28]. Little is known about mental healthcare professionals’ and users’ perspectives on how to address these problems. A key challenge for hospital mental healthcare services is the lack of clarity on what constitutes coherence in intersectoral care between hospital mental healthcare and community mental healthcare [21,29]. The limited knowledge regarding the coherent intersectoral care between hospital mental healthcare and community mental healthcare shows a gap in research. We therefore need more empirical research on the healthcare professionals’ and users’ experience of coherent intersectoral care in order to be able to provide higher-quality care within the intersectoral sector [21,24,27,30,31,32].

The aim of this study is to explore coherency in intersectoral care between hospital mental healthcare (Hospital mental healthcare includes in- and outpatients) and community (Community mental healthcare is addressed as a municipal social psychiatric service) mental healthcare, as experienced by healthcare professionals and users.

## 2. Materials and Methods

The method of analysis is inspired by French philosopher Paul Ricoeur, in which the structure in the analysis is the narrative theory about mimesis divided into three; naive reading, structural analyses, and critical interpretation. The study’s epistemology was framed by a qualitative approach using Ricoeur’s phenomenological-hermeneutic theory of interpretation for healthcare professionals’ and users’ experience of coherent intersectoral care. The unique contribution from this investigation is part of a study where the data were used to perform iterative analyses with different theoretical framings in order to improve methods and depth by triangulation [29].

### 2.1. Recruitment and Sampling

We used purposive sampling [32,33] to establish contacts and recruit participants for this study. Once we had received the names of the participants from the managements of hospital mental healthcare, Recovery College, community mental healthcare, and user associations, the first and second authors contacted the participants by email or telephone to request an appointment. The time period for the study was established as 2016 to 2019, where all of the authors were colleagues in the nurse education program. The first author holds a Ph.D. degree and has several years of theoretical and clinical experience in mental health; the second author is a graduate with no previous experience in psychiatry; and the third author holds a Ph.D. degree and has project experience, but not mental health experience.

As the research group, we reflected upon and discussed our positions in the research field, as well as our experiences.

As participants, the mental healthcare professionals (nurses, social workers, educators, and physiotherapists) were employed in either hospital mental healthcare or community mental healthcare. In total, 34 informants gave their informed consent to participate (Table 1). Of these, 14 represented a user perspective, while 20 were healthcare professionals, of which 12 were from hospital mental healthcare and eight from the community mental healthcare.

The healthcare professionals had 2–25 years of experience in mental healthcare, most having lengthy experience as specialists within the field of mental healthcare and employed for more than 5 years.

The users had different types of challenges and diagnoses (although mainly schizophrenia) and had experience using the services in both sectors. Most lacked an occupation, but several were active in volunteering.

Users with experience of coherent intersectoral care in both sectors (hospital mental healthcare and community mental healthcare) were included, and the healthcare professionals included had previous experience in the delivery of intersectoral care.

Users with distinctly psychotic and paranoid episodes lacking the energy or desire to participate were excluded. We have chosen not to involve the relatives in the study.

### 2.2. Data Collection—Focus Group Interviews

Focus group interviews (FGIs) were conducted, which aimed to discuss and gain a deeper understanding of the findings. Data were collected through nine FGIs. A key strength of a FGI is that it encourages interaction in the group, which creates new perspectives and a deepened discussion [33]. The participants in the focus groups were unfamiliar with each other and were selected because they had certain characteristics in common that related to the topic of the focus group. Within FGIs, the moderator and interviewer create a permissive and nurturing environment that encourages different perceptions and points of view, without pressuring participants to vote, plan, or reach a consensus [34].

Focus groups with 4–8 participants provide an opportunity for dynamic and mutual inspiration, which can make the empirical data more comprehensive [34]. Some users joined the group for a cup of coffee, but left shortly afterwards as they were not due to participate. The participants in the focus group were very talkative.

All FGIs were audio-recorded and transcribed verbatim [35]. Inspired by existing knowledge in the field [8,28], we formulated a list of topics to explore our participants’ experience of coherent intersectoral care within mental healthcare; these themes guided the narrative. Based on the topics, we asked a few broad questions [31] and focused on follow-up questions to obtain richness and depth (Table 2). The data have previously been used in a published document analysis, where we explored the discourses that emerge when healthcare professionals and users relate to intersectoral care [17].

Two interviewers were present at each FGI interview in order to ensure a dialogic conversation where narrative phenomena were disclosed and developed. The first author was present in most FGIs, which reinforced the role of interviewer. There was no prior relationship between researchers and participants. The Consolidated Criteria for Reporting Qualitative Research checklist was used as a guideline to ensure accurate and complete reporting of the study [36]. This guide contains recommendations for qualitative research, which we have tried to achieve throughout.

The topics served as a guiding framework for the knowledge we wanted to extract, but in the FGIs, we let ourselves flow with the thoughts voiced by the participants and inquired more deeply into these to achieve a nuanced understanding of their opinions [33].

### 2.3. Data Analysis

The aim of a hermeneutic interpretation of a text is to understand the participants’ experiences as revealed from the text itself. We interpreted what the text says and speaks about [30]. Aiming to find a deeper understanding and for creating new knowledge, FGIs were explored through descriptions in FGIs, where the interpretation already begun.

The analysis was carried out by all authors inspired by the phenomenological-hermeneutic approach described by Ricoeur’s theory of text interpretation, by following three levels of interpretation: naive reading, structural analysis, and critical interpretation.

The FGIs were analyzed individually. Once the analysis of the narrative interviews was completed, we then analyzed across all of the FGIs. The naive reading gave an overall impression of what had been said. In the naive reading, the text was read repeatedly and with an open mind to achieve an initial understanding of how users experienced coherent intersectoral care. The units of opinion can be part of a sentence or several sentences, and it is important to emphasize that, in all cases, they can express only one opinion.

A structural analysis is a step between the naïve surface interpretation and comprehensive understanding [37,38]. In structural analysis, units from the naive reading were used to find out what had been discussed and helped to formulate units of significance and corresponding themes. In the structural analysis of the FGIs, the focus was on deepening our understanding of the structures and patterns in the texts. To explain the structures and patterns in the FGIs, we examined whole, multiple, or parts of sentences from the text. Structural analyses were performed several times until we came to the point where no further subthemes and themes could be found in the data. The structural analysis aims to develop a possible explanation about the text, thus opening up the text and enabling the researcher to reach a deeper and more critical interpretation. This is situated at the end of the hermeneutic arc, thus creating a comprehensive interpretation of the text. The analysis is based on the total material and consists of the ‘data’, ‘text’, ‘condensation’, ‘subtheme’, and ‘theme’ phases (Table 3).

The comprehensive understanding emerged from our reading and rereading, checking and rechecking, until we had reached a common understanding of the phenomena [29]. The themes from the structural analysis became the basis of the comprehensive understanding and were presented in the discussion, where theoretical perspectives were brought forward to achieve new insight and gain new knowledge regarding coherent intersectoral care.

### 2.4. Ethical Considerations

This study was conducted in adherence to the ethics of scientific work. Informed consent was obtained from all participants after they had received verbal and written information regarding the purpose of the study. The participants were informed that they were able to withdraw from the study with no consequences at any time, and that all data would be treated in such a way that no unauthorized person could have access to the material. Data were treated confidentially and anonymously, and all data material will be destroyed following publication [39].

We adhered closely to the ethics of scientific work. The study was accepted by the Danish Data Protection Agency According to the Helsinki Declaration [40] and Danish law [41], no formal permit from a biomedical ethics committee was required, as the purpose of the research was not to influence the informants, either physically or psychologically.

## 3. Findings

Initially, we first analyzed the healthcare professionals’ experience, followed by the users’. The analysis of themes is supported by citations from the FGIs’ data material.

### 3.1. Common Language and Culture: Experience of Healthcare Professionals

The healthcare professionals’ experience of coherence in intersectoral care was coded into one main theme: ‘common language and culture’. The main theme was based on three themes and nine subthemes. The first theme was ‘coherence in intersectoral care involves good internal and external collaboration’ with its three subthemes ‘both sectors are located in close proximity’, ‘treatment and rehabilitation are based on a recovery-oriented approach’, and ‘contact person in the coherent intersectoral care’. The next theme was the ‘healthcare professionals’ focus on the methods to create a recovery-oriented approach to the intersectoral care’ with its two subthemes ‘cognitive behavioral therapy’ and ‘the users often do not fit the treatment methods’. The third theme was ‘barriers to successful intersectoral care’ with its four subthemes ‘lack of knowledge of the other employees in the cooperating intersectoral units’, ‘healthcare professionals do not understand medicine and are afraid of patients’, ‘users must be in a life-threatening condition before the mental health center will activate a treatment judgment’, and ‘frequent change of employees hurts ‘know-how’’ (Table 4).

### 3.2. Coherence in Intersectoral Care Involves Good Internal and External Collaboration

Under the above theme, three subthemes were unfolded: ‘both sectors are located in close proximity’, ‘treatment and rehabilitation are based on a recovery-oriented approach’, ‘contact person in the intersectoral care’, as presented below.

#### 3.2.1. Both Sectors Are Located in Close Proximity

As experienced by healthcare professionals, intersectoral care was most coherent when the units were located in close proximity. ‘We are an outpatient clinic and operate close to an inpatient ward. So, we visit them and have a slightly easier communication path’ (Professional FG 1).

#### 3.2.2. Treatment and Rehabilitation Are Based on a Recovery-Oriented Approach

The intersectoral collaboration was experienced as recovery-oriented, and the focus was always on the users’ needs and hopes.

‘The intersectoral care is very recovery-oriented. When users are discharged, the healthcare professionals in community mental healthcare support in the home’ (Professional FG 1). ‘We have the opportunity to reach out to users on the first visit, if appropriate and necessary’ (Professional FG 3).

Various methods were described by the healthcare professionals to promote recovery-oriented intersectoral care. Tools that could be used to coordinate and create transparency in healthcare professionals acting across sectors were emphasized as part of the solution. Network meetings, weekly and treatment plans were among these tools.

‘We experience that the approach to coordinate coherent intersectoral care was through network meetings, weekly and treatment plans as methods to keep the focus on the user’s recovery’ (Professional FG 2). ‘There are many parts of the sector transition. In our work, it is very much about coordination’ (Professional FG 5).

#### 3.2.3. Contact Person in the Coherent Intersectoral Care

The contact person in this coherent intersectoral care was experienced as being important to ensure continuity, and, as an organizational structure, as one of the best ways to build bridges between the sectors and to ensure cohesion. ‘The contacts we have are the ones who coordinate. They are the ones who follow up appointments and also follow-up to ensure that we get to hold those meetings and medical appointments’ (Professional FG 3).

The healthcare professionals focus on the methods to create a recovery-oriented approach to the coherent intersectoral care.

In the theme below, the healthcare professionals reflected on different structural methods as recovery-oriented. From this perspective, their methods were reflected as approaches to promoting coherent intersectoral care. The healthcare professionals’ focus on coherent intersectoral care was formulated into two subthemes; ‘cognitive behavioral therapy’ and ‘the users often do not fit the treatment methods’, as presented below.

#### 3.2.4. Cognitive Behavioral Therapy

The healthcare professionals experience cognitive behavioral therapy as a framework for coherent intersectoral care in hospital mental healthcare. The healthcare professionals in hospital mental healthcare were convinced that the use of cognitive behavioral therapy was the best basis for a recovery-oriented practice. By using these methods, the users learned about their illnesses and symptoms and were empowered to manage their conditions.

‘The cognitive model is the way we have a conversation, the way we structure agreements. We have an agenda, what has happened lately, what topics should we be dealing with, what home tasks, and what medicine’ (Professional FG 2).

#### 3.2.5. The Users Often Do Not Fit the Treatment Methods

According to the healthcare professionals in the hospital mental healthcare, the users did not always fit the treatment methods. Many users with severe mental illnesses found it difficult to reflect on their situation and to answer all the questions within the cognitive behavioral therapy model.

‘Users do not have a completely realistic understanding of what their share is in working with cognitive behavioral therapy, where I think that, in psychiatry. I think it is unbearable and too difficult for everyone’ (Professional FG 2).

### 3.3. Barriers to Successful Coherent Intersectoral Care

Several barriers were found to creating a successful and coherent intersectoral care process between hospital mental healthcare and community mental healthcare. These barriers were categorized into four subthemes: ‘lack of knowledge of the other employees in the cooperating intersectoral units’, ‘healthcare professionals do not understand medicine and are afraid of patients’, ‘the users must be in a life-threatening condition before the hospital mental healthcare activates a treatment judgment’ and ‘frequent change of employees hurts know-how,’ as presented below.

#### 3.3.1. Lack of Knowledge of the Other Employees in Coherent Intersectoral Care

The healthcare professionals experience a lack of knowledge about the employees in the collaborating intersectoral service units. The healthcare professionals did not know the employees of the other service units, and thus communication and collaboration between intersectoral employees becomes difficult.

‘We have a lack of knowledge about each other’s workflows and structures. They will probably not be happy when they hear that they will receive the patient tomorrow because they need days to prepare’ (Professional FG 1).

‘Physically, it is an advantage that we as an ambulatory section live right on top of the other wards, as we can visit inpatient users below. That gives an easier communication path. It creates consistency and confidence for the user’ (Professional FG 1).

#### 3.3.2. Healthcare Professionals Do Not Understand Medicine and Are Afraid of Patients

Healthcare professionals in the hospital mental healthcare claimed that healthcare professionals in community mental healthcare did not understand medicine and were afraid of the users. In community mental healthcare, healthcare professionals were predominantly focused on pedagogical activities. The following comment of a healthcare professional in mental healthcare illustrates the lack of mutual understanding of the users’ problems and who should help solve them:

‘At one time, we held a dialogue with the healthcare professionals in community mental healthcare who were sending users to the hospital’s mental healthcare. We could not see the purpose when the user was the same as he was back when he was discharged from the hospital’s mental healthcare department. The healthcare professionals at community mental healthcare claimed he could not stay in the institution because he was too ill. Sometimes healthcare professionals arrive at the mental healthcare with a user, but the user will absolutely not be admitted’ (Professional FG 1).

#### 3.3.3. The Users Must Be in a Life-Threatening Condition before the Hospital Mental Healthcare Activates a Treatment Judgment

The healthcare professionals from community mental healthcare experienced that healthcare professionals in hospital mental healthcare did not take them seriously. When they informed the hospital mental healthcare that a user needs to be admitted, the user was often rejected or quickly discharged. Coherence in intersectoral care is challenged by a lack of collaboration due to the different cultures; healthcare professionals in community mental healthcare could assess the user as, for example, violent and psychotic and in need of admission, while healthcare professionals in the hospital mental healthcare do not believe there is any need for hospitalization. The healthcare professionals in community mental healthcare did not feel listened to and felt that they were powerless when they view a user as needing acute treatment but healthcare professionals in the hospital mental healthcare would not accept the user. Users should be in a life-threatening condition before the hospital mental healthcare activate a treatment judgment (this is a requirement for legislated treatment):

‘Our collaboration with hospital mental healthcare is challenged when a user does not take his medicine, and a treatment judgment is made. When we call the ward to activate the judgment, it will often be rejected on the grounds that we have overreacted’ (Professional FG 5).

#### 3.3.4. Frequent Change of Employees Diminishes Understanding

Frequent change of healthcare professionals is detrimental to the establishment of ‘know-how’ and was experienced as a serious problem because the healthcare professionals and users did not understand the work routines and schedules of their colleagues in intersectoral care, and therefore feel that they have to constantly restart the collaboration. The coordination during the intersectoral transition between the two sectors was challenged by frequent changes of physicians and other healthcare professionals:

‘Frequent change of healthcare professionals diminishes understanding. When I call, they are no longer there. The expression of the condition is the same all over in healthcare service’ (Professional FG 4).

### 3.4. Not Experiencing Being Seen as an Individual Person: Experience of the Users

In the FGIs, the users’ experiences of intersectoral care were reflected in one main theme: ‘users did not experience being seen as an individual person’. The users did not experience that the healthcare professionals focus on individual needs, but rather that the users have to adapt to the professionally preferred mode of treatment and rehabilitation. The main theme was based on two themes and five subthemes found in the analysis. The first theme was ‘limited opportunity to participate in their treatment and rehabilitation’ with its two subthemes ‘lack of information about transitions to other units’ and ‘hospitalization starts with a conversation about the discharge’. The next theme was ‘the users did not participate much in their treatment and rehabilitation’ with its three subthemes ‘not feeling respect for lived experience’, ‘not invited to meetings where the treatment and rehabilitation is planned’, and ‘not listening to the user’s individual wishes to the treatment and rehabilitation’ as presented below (Table 5).

### 3.5. Limited Opportunity to Participate in Their Treatment and Rehabilitation

This theme reflects how the creation of coherence in intersectoral care was largely dependent on the user’s will and ability to take responsibility for information moving from one unit to another. Coherence in intersectoral care requires that the user has resources to participate and collaborate actively

‘I am surprised at the lack of influence on the process. Also compared to when I felt ready. The two treatment sites did not know what each other was doing. I think that is very strange and unsafe. The context is left to me’ (User FG 9).

#### 3.5.1. Lack of Information about Transitions to Other Units

Lack of information about transitions to other units means that the user’s treatment plans did not follow him, and he must start all over again every time he gets to a new unit in mental healthcare: ‘I have been in the hospital’s district psychiatry and community mental healthcare for a while, and my experience is that through every transition, it’s like starting over anew. The ideal would be someone who could follow me. And how should information follow me? There is probably a journal somewhere’ (User FG 8).

#### 3.5.2. Hospitalization Starts with a Conversation about the Discharge

As experienced by the users, admission starts with a conversation about the discharge. The users found admission to be very stressful when the healthcare professionals start the conversation with a discussion about when he must be discharged. ‘One thing some healthcare professionals say when you are admitted is that in fourteen days or three weeks, you will come out again. So, I say, how do they know? So, I tell them that it’s of no use to me, when they give me a date for my discharge. Because I don’t know how I got there’ (User FG 6).

The above citation could relate to a bigger problem: Many users did not feel ready to be discharged shortly after admission. The focus was on efficiency and rapid treatment so that users could move forward as quickly as possible.

### 3.6. The Users Did Not Participate Much in Their Treatment and Rehabilitation

The users did not participate much in their treatment and rehabilitation, while they did not feel any respect for lived experience, were not invited to meetings where the treatment and rehabilitation is planned, and did not feel their individual wishes for the treatment and rehabilitation were listened to.

Many users did not have the resources to be active and participate when they were admitted with serious illnesses. This problem was a general one across the sectors where healthcare professionals expected the users to follow the plan and participate actively. The healthcare professionals steered the plans for the treatment and rehabilitation. For many users, the experience was that participation was not possible to the extent that they had hoped for. The users did not experience that they could participate much in decision-making. ‘I would like to be involved a little more in the treatment. Now, I didn’t know that I should be in here for two-and-a-half months, and there was probably no one here who knew anything about it. But I would like to think that once a month had passed, you can say: “Okay, now a month has passed, it has happened and that and that. We have these problems and these thoughts going forward with the medicine. For example, we have put your medicine down here, or we want to put your medicine up”. And if they, for example, wanted to put up my medicine, then I also want to be involved in it. And I wasn’t’ (User FG 9).

#### 3.6.1. Not Feeling Respect for Lived Experience

Users felt that their life experience was not being respected, which we interpreted as the users attempting to share their experiences of suffering and needing help but not having those needs met: ‘For me, a good patient process is when you arrive at, for example, the district psychiatrist’s outpatient treatment facilities, that they actually know who you are when you arrive. Not necessarily your entire life story, but at least they know your name. They know what your diagnosis is, they are aware of what problems you are currently battling with’ (User FG 6).

#### 3.6.2. Not being Invited to Meetings where the Treatment and Rehabilitation Is Planned

Not being invited to meetings where treatment plans are conducted was a significant problem for users who had wishes and hopes for the future, and who hoped to discuss these details with the healthcare professionals. Moreover, the users did not receive any information about why they should not attend the network meetings. ‘I have not attended network meetings and my experience is that the meetings are mostly held for the sake of the healthcare professionals rather than listening to us users’ (User FG 5).

#### 3.6.3. Not Listening to the Users’ Individual Wishes to the Treatment and Rehabilitation

Not listening to the users’ individual wishes for the future was an experience that users problematized. We interpreted this as the users wishing that the intersectoral collaboration was more focused on support their recovery process. ‘When it comes to purely medical treatment, you have good contact with each other. But when it comes to physiotherapy, occupational therapy and that path, it is as if there is no communication at all’ (User FG 7).

## 4. Discussion

In the analysis, we interpreted that healthcare professionals and users show different perceptions and experiences of coherent intersectoral care between hospital mental healthcare and community mental healthcare. In the analysis, we interpreted that healthcare professionals and users show different perceptions and experiences of coherent intersectoral care between hospital mental healthcare and community mental healthcare. There are differences between the professionals’ and users’ perception of ‘coherence’. While the professionals relate closely to their own practice and context, users perceive coherence related to not just continuity between sectors and easy handovers. Coherence also seems to involve the integration of the psychiatric services/help in user’s daily life. The overriding theme from a healthcare professional perspective ‘common language and culture’ (the interpretation of which differs between the sectors) and the theme from the users’ perspective ‘not experiencing being seen as an individual person’, is seen as opening up the discussion for the different ways in which healthcare professionals understand and solve users’ problems.

Overall, we interpreted that the healthcare professionals primarily focus on the treatment of symptoms and of fostering self-care and rely less on involving the user. The healthcare professionals experience the treatment and rehabilitation in intersectoral care as being recovery-oriented, but their focus was framed by clinical recovery where the goal is remission and maintenance of the functional level, as we also see among other researchers [42,43,44]. The healthcare professionals in community mental healthcare thought that relational work is the most effective way to help the users towards recovery.

In addition, other studies show that healthcare professionals in hospital mental healthcare are focused on disorders and symptoms and do not prioritize the involvement of users in the treatment and rehabilitation [44,45]. The lack of a common language and culture becomes a barrier to improving coherence in intersectoral care, as is also shown in other studies [24,30]. In addition, frequent replacement of healthcare professionals, meagre economic resources and low numbers of healthcare professionals, all make the transition between hospital mental healthcare and community mental healthcare difficult; all of these issues are detrimental to the creation and maintenance of a shared understanding.

Healthcare professionals in community mental healthcare have voiced that plans prepared for continued mental healthcare after discharge from hospital mental healthcare do not necessarily make sense for the users.

As Bonfils et al. emphasize, differences in language and culture are visible through the healthcare professionals’ different views about when a person is ill and in need of help [5,20.26]. Coherent intersectoral care requires the communication of essential information to be given in a way that other units understand and accept [24,45]. We believe, in accordance with [42], that there is need for a cultural change, placing users at the center of their own recovery process. Therefore, it is essential that coherent intersectoral care supports the users’ need for knowledge of care rather than following predefined treatment procedures that do not necessarily match the users’ real needs. Karlsson and Borg believe that we should rethink the healthcare system so that it emphasizes the differences between individual users, rather than seeing them as a collective mass [42].

In this and other studies, healthcare professionals agreed that the users should participate in their treatment and rehabilitation, but, because of their severe symptoms, it often seems difficult to involve them [10,16]. In this and other studies, users with serious symptoms were not invited to participate in planning the process of help, and the healthcare professionals thus decided the users’ treatment without involving or informing them, as they expected this group of users to be unable to fully participate [10,11]. The consequence of such expectations is that the users do not gain ownership of a treatment that should address their individual needs.

In hospital mental healthcare, there is confidence that methods such as psychoeducation and cognitive behavioral therapy provide a coherent fostering basis for intersectoral care and collaboration [16,19]. According to healthcare professionals, when the users gain more knowledge and ownership of their treatment, the recovery-oriented process is experienced as being more coherent. The municipality contributes on many fronts and has the expectation that all users will, as far as possible, recover in order to be able to care for themselves at some point. Users are thus subject to the respective plans formulated by the hospital mental healthcare and community mental healthcare, but their participation in the preparation of these plans (often) occurs only in the most peripheral sense.

### Methodological Considerations

Ricoeur’s hermeneutics is a challenging and time-consuming method that has led to in-depth discussions in the research team in order to arrive at relevant themes. As consequence of the phenomenological-hermeneutic methodology, we deliberately chose a small number of participants for this study. We found the data collected to be adequate with respect to our opportunities to perform the thorough analysis that is required by the phenomenological-hermeneutic tradition, as well as sufficient to formulate well-founded themes and main themes from the interpretation of the texts from FGIs with users and healthcare professionals.

The findings from this study have implications for practice and research. The healthcare professionals can learn from the healthcare professionals’ and users’ experience of intersectoral care and collaboration. Our findings may provide greater insight into the obstacles that may be associated with creating a link between intersectoral care between hospital and community mental healthcare.

The healthcare professionals clearly have a central role in the promotion of coherent intersectoral care. Reflection on their own practice may make healthcare professionals aware of their current approach towards engaging users, which can lead to nurses acknowledging the patients’ role in nursing care and a more patient-centered manner.

We advocate for the relevance of new research seeking to improve continuity and coherence for users’ recovery process by developing a user-participated intervention drawing on users’ hopes, goals and possibilities. This will require a discussion on how successful transitions between hospital mental healthcare and community mental healthcare should be understood and implemented in the nursing practice. Anchoring the transition between hospital mental healthcare and community mental healthcare requires changes in organizational structures and procedures.

## 5. Conclusions

The aim of this study was to explore coherency in intersectoral care between hospital mental healthcare and community mental healthcare, as experienced by healthcare professionals and users. The study shows that healthcare professionals from both sectors experience that coherent intersectoral care was defined by the time in which users received treatment and/or rehabilitation. The healthcare professionals experience that intersectoral care was dependent on the intersectoral collaboration and entails a recovery-oriented focus, where users were involved in decisions, and that communication was optimized when the healthcare professionals across the sectors knew each other in advance and the units were physically close. Neither healthcare professionals nor users experience the intersectoral care and collaboration as a coherent whole because information was often lost in the transitions between sectors, as well as in the structural conditions such as network meetings where user-involvement in decision-making was not complied with. This leads us to understand the experience of intersectoral care as an ideal often lost between sectors.

The healthcare professionals in hospital mental healthcare and community mental healthcare are divided on the question of how the users’ problems should be understood and resolved. Different paradigmatic approaches (e.g., biomedicine and holism [43,44] seem to challenge what should be the focus of the users’ recovery process between hospital mental healthcare and community mental healthcare. The users had no doubt that they want to be seen more as whole persons and were demanding a more holistic paradigm across the sectors. The users did not experience being sufficiently involved in the intersectoral care and being seen as a whole person with a life outside hospital mental healthcare. The users did not feel the professionals listened to their perceptions of mental challenges and their narratives about their lives were not passed on between treatment sites.

## Figures and Tables

**Table 1 ijerph-17-06510-t001:** Focus groups, context, and participants.

Focus Groups	Context	Participants
**1.**	Hospital mental healthcare	N = 6 Healthcare professionals
**2.**	Hospital mental healthcare	N = 3 Healthcare professionals
**3.**	Hospital mental healthcare	N = 3 Healthcare professionals
**4.**	Community mental healthcare	N = 4 Healthcare professionals
**5.**	Community mental healthcare	N = 4 Healthcare professionals
**6.**	Recovery college	N = 3 Users
**7.**	Community mental healthcare	N = 4 Users
**8.**	Hospital mental healthcare	N = 4 Users
**9.**	User organization	N = 3 Users
	Number of participants	34

**Table 2 ijerph-17-06510-t002:** Interview guide.

Topics
Overall experience of intersectoral care between hospital mental healthcare and community mental healthcare
Examples of intersectoral care Patient participation Confidence
Communication Relatives
Continuity and coherence Small summary
Continuity Transitions Individual focused
Opinion/Confidence/Relationship
Continuity Transitions Individual focused
Opinion/Confidence/Relationship

**Table 3 ijerph-17-06510-t003:** Example of structural analysis of narrative interviews with healthcare professionals and users.

Data	Text	Condensation	Subtheme	Theme
FGI: Healthcare professional hospital mental healthcare	Physically, it is an advantage that we operate right by the wards. We can contact the employees and there is close dialogue. This creates coherence	Communication breaks down barriers.	The units are located in close proximity.	Coherence in intersectoral care involves good internal and external collaboration.
FGI: Health professional in community mental healthcare	The community mental healthcare draws up a plan for the users that defines the focus of the intersectoral care; that is to say, the aim to be achieved by the users. Some users do not agree with the plans and they do not attend the meetings.	The intersectoral care is decided by the healthcare professional in hospital mental healthcare and community mental healthcare.	Municipal Requirement.	Barriers to good intersectoral care.
FGI: Users	The day after I arrived at Gentofte hospital’s mental healthcare department, I was admitted to a closed ward. I did not really get any information about why I should be in the closed ward. I was quite confused about this. And then the day after I was moved to the open ward.	I did not feel consulted or involved.	Lack of information about transitions to the other wards. Discharge and intersectoral care is planned by the healthcare professionals, but not necessarily for the users.	Healthcare professionals are steering intersectoral care.

**Table 4 ijerph-17-06510-t004:** Intersectoral care as experienced by healthcare professionals: subthemes, themes, and main theme.

Subthemes	Themes	Main Theme
Both sectors are located in close proximity. Treatment and rehabilitation are based on a recovery-oriented approach. Contact person in the intersectoral care.	Coherent intersectoral care involves good internal and external collaboration	Common language and culture
Cognitive behavioral therapy. The users often do not fit the treatment methods.	The healthcare professionals focus on the methods to create a recovery-oriented approach to intersectoral care.	
Lack of knowledge of the other employees in coherent intersectoral care. Healthcare professionals do not understand medication and are afraid of patients. The users must be in a life-threatening condition before the hospital mental healthcare applies a treatment judgment. Frequent change of employees diminishes understanding.	Barriers to successful intersectoral care.	

**Table 5 ijerph-17-06510-t005:** Intersectoral care as perceived by users: subthemes, themes, and main theme.

Subthemes	Themes	Main Theme
Lack of information about transitions to other units. Hospitalization starts with a conversation about the discharge.	Limited opportunity to participate in their treatment and rehabilitation.	Not experiencing being seen as an individual person: experience of the users.
Not feeling respect for lived experienceNot invited to meetings where the treatment and rehabilitation is planned. Not listening to the users’ individual wishes regarding the treatment and rehabilitation.	The users did not participate much in their treatment and rehabilitation.

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
