# Peer review of "Healthcare Professionals’ and Users’ Experiences of Intersectoral Care between Hospital and Community Mental Healthcare"

_ijerph, 2020, doi:10.3390/ijerph17186510_

Round 1

Reviewer 1 Report

Are the methods adequately described?

Is it possible to elaborate more on focus group interviews (p 119-140)?

Shortly to explain your specific use of FGI? Do you have any reflections on your own role as interviewers? How did you facilitate or moderate* the nine FGI groups?

Are there any challenges concerning validity by using FGI in your research?

* Bente Halkier (p 143) in "Kvalitative metoder" edited by Svend Brinkmann & Lene Tanggaard, Hans Reitzels 2015.

Author Response

Dear reviewer

Thank you very much for your constructive comments.

We have added more information on focus group interview and hope it meets your expectations.

We have revised this reference: Bente Halkier (p 143) in "Kvalitative metoder" edited by Svend Brinkmann & Lene Tanggaard, Hans Reitzels 2015.

Best regards 

the authors 

Reviewer 2 Report

The title of the article is not easy to comprehend. Please consider a more accurate and readable title. The title should be a teaser and let the reader know what’s to come.

The article brings focus on barriers to the collaboration within psychiatry, but it is unclear what kind of services/units the article refers to. Units within psychiatric hospitals are far from similar and community mental healthcare vary deeply from one country to another. Please be specific. That is, what does the term ‘community mental healthcare’ actual refer to? Supported houses? Local primary care services? Self-help groups for mental health? Or?

Line 41 – challenges and criticisms of the mental healthcare system is due to a range of factors – lack of coherent intersectoral process is just one, please soften your point.

Line 42 – “The Danish Society for Quality in the Health Sector”. Is this an official title or your translation? Please put in a reference.

Line 44/45 – the article states that “intersectoral collaboration is an important tool in order to achieve coherent”. And in the introduction – line 78 – it is stated: “The aim of this study is to explore coherency”. What does the term ‘coherent/coherency’ actual refer to in this article? Do the authors have a preconceived idea of ‘coherent/coherency’? Or, does the study try to explore how and why (and if) ‘coherent/coherency’ is important to the informants? Please elaborate in the introduction.

Line 54 – reference is needed.

Line 74 – yes, there is a gap in research. But there are plenty of documents - evaluations and rapports ('grey literature') - on the subject – from a Danish context. What do we learn from these? See e.g. https://www.sdu.dk/da/sif/rapporter/2017/sammenhaeng_i_indsatsen_for_mennesker_med_psykiske_lidelser.

Line 81- Paul Ricoeurs theory. Please describe (in your own words) what the theory consists in and how it is of use to your data-material (just short).

Line 118 - Focus group interview – how did they go? Again – just short, were the informants talkative? Did they have a lot on their mind regarding your questions or … ?

Line 119 – which initial findings?

Line 129 – reference is needed.

Line 131 – table 2. No need for a table. Just describe the interview themes or questions. It seems as an attempt to present qualitative data by quantitative standards.

Line 136 – Regarding the ‘Consolidated Criteria for Reporting Qualitative Research checklist'. Please describe (in your own words) this list and how it is of use to exactly your study. Also, a reference is needed.

Line 188 (3. Findings) - The presentation of the results/findings are very much in headlines (main theme, themes and subthemes) – and these are repeated several times. First in the actual text, then in a table and then in the text again. This leaves rather little room for unfolding the actual empirical material. Again - it almost seems as an attempt to present qualitative data by quantitative standards. This might be a question of style regarding the presentation of empirical material, though. Consider less headlines and offer more insight into the empirical material.

Line 174 – which theoretical perspectives?

Line 183 – which ones? Be more specific regarding ethics. What is especially relevant for your study?

Line 362-363 – how is this ‘comment’ relevant? Does it relate to the question of coherency, or?

Line 411+412. The article states that the data opens for at discussion of the different ways professionals and users understand and solve user’s problems. That’s true. Maybe it also opens for a discussion of professional and users experience and perception of ‘coherency’. Is there a difference? It seems (from your data) that a coherent system/coherent services from a professional perspective and coherency for users are not quite the same. For users, coherency seem to be related to not only continuity between sectors and easy handovers. Coherency also seems to involve the integration of the psychiatric services/help in user’s daily life (where most of people’s recovery unfold). So again – do the authors have a preconceived idea of ‘coherent/coherency’? Does the study (also) try to explore the very experience and perception of coherency? This would be interesting and relevant. Please specify.

Line 454 - Methodological considerations – is mainly about implication for practice and research – and does not consider methods much.

Line 490 – “paradigmatic approaches (biomedicine and holism…”. This is a rather simplified description – a reference is needed, at least.

I do not feel qualified to judge about the English language, but there are sentences that seems to need some kind of editing, because it is simple hard to know what the authors actually mean, e.g. line 90: “ The time in the study was established in a research group …” or line 152: “The units of opinion can be part of…” Look the text carefully through, please.

The conclusion hardly sums up users experience and perception of coherency. Please do.

Author Response

Dear reviewer

Thank you very much for constructive comment. 

The title of the article is not easy to comprehend. Please consider a more accurate and readable title. The title should be a teaser and let the reader know what’s to come.

- We have shortened the title so that it appears more clear.

The article brings focus on barriers to the collaboration within psychiatry, but it is unclear what kind of services/units the article refers to. Units within psychiatric hospitals are far from similar and community mental healthcare vary deeply from one country to another. Please be specific. That is, what does the term ‘community mental healthcare’ actual refer to? Supported houses? Local primary care services? Self-help groups for mental health? Or?

  • We completely agree and have tried to address this more clearly.

Line 41 – challenges and criticisms of the mental healthcare system is due to a range of factors – lack of coherent intersectoral process is just one, please soften your point.

  • We have softened the sentence.

Line 42 – “The Danish Society for Quality in the Health Sector”. Is this an official title or your translation? Please put in a reference.

- Thank you, we have now put in a reference. 

Line 44/45 – the article states that “intersectoral collaboration is an important tool in order to achieve coherent”. And in the introduction – line 78 – it is stated: “The aim of this study is to explore coherency”. What does the term ‘coherent/coherency’ actual refer to in this article? Do the authors have a preconceived idea of ‘coherent/coherency’? Or, does the study try to explore how and why (and if) ‘coherent/coherency’ is important to the informants? Please elaborate in the introduction.

-This is an important issue that we have now tried to address. We want to explore intersectoral collaboration to achieve coherence between the two sectors. We have now clarified that some studies find intersectoral collaboration is an important tool in order to achieve coherence and that this is what we will explore further.

Line 54 – reference is needed.

- we have add it 

Line 74 – yes, there is a gap in research. But there are plenty of documents - evaluations and rapports ('grey literature') - on the subject – from a Danish context. What do we learn from these? See e.g. https://www.sdu.dk/da/sif/rapporter/2017/sammenhaeng_i_indsatsen_for_mennesker_med_psykiske_lidelser.

- Many thanks for the link to the SDU report. We have added considerations from gray literature. The text has been moved due to corrections to line 118.

Line 81- Paul Ricoeurs theory. Please describe (in your own words) what the theory consists in and how it is of use to your data-material (just short).

- We have tried to accommodate this (Now line 139-142). 

Line 118 - Focus group interview – how did they go? Again – just short, were the informants talkative? Did they have a lot on their mind regarding your questions or … ?

Some users showed up for the interview, join the group, had a cup of coffee and ate cake, but left shortly after as they would not attend anyway.

The participants in the focus group were very talkative.

Line 119 – which initial findings?

We have change it to 'the findings' 

Line 129 – reference is needed.

We have now add it. see now line 153. 

Line 131 – table 2. No need for a table. Just describe the interview themes or questions. It seems as an attempt to present qualitative data by quantitative standards.

we have in several publications experienced that reviewers and editors demand that kind of table. We are mostly in favor of preserving it, but if the editor also thinks it should be removed, we of course agree.

Line 136 – Regarding the ‘Consolidated Criteria for Reporting Qualitative Research checklist'. Please describe (in your own words) this list and how it is of use to exactly your study. Also, a reference is needed.

Reference is now added, and add text about this checklist. Now line 163-164. 

Line 188 (3. Findings) - The presentation of the results/findings are very much in headlines (main theme, themes and subthemes) – and these are repeated several times. First in the actual text, then in a table and then in the text again. This leaves rather little room for unfolding the actual empirical material. Again - it almost seems as an attempt to present qualitative data by quantitative standards. This might be a question of style regarding the presentation of empirical material, though. Consider less headlines and offer more insight into the empirical material.

now line 216. We do not entirely agree with this criticism. We think it creates a clear structure, where the reader is continuously informed in relation to each theme and where the table supports all the findings. We hope it can be accepted.

Line 174 – which theoretical perspectives?

We have addressed to:The analysis was carried out by all authors inspired by the phenomenological–hermeneutic approach described by Ricoeur’s theory of text interpretation, by following three levels of interpretation: naive reading, structural analysis and critical interpretation.

Line 183 – which ones? Be more specific regarding ethics. What is especially relevant for your study?

We have deepened our ethical considerations.

Line 362-363 – how is this ‘comment’ relevant? Does it relate to the question of coherency, or?

It's a quote from a user. It shows something about the expectations users are met with during an admission.

‘One thing some healthcare professionals say when you are admitted is that in fourteen days or three weeks, you will come out again. So, I say, how do they know? So, I tell them that it’s of no use to me, when they give me a date for my discharge. Because I don't know how I got there’ (User FG 6).

Line 411+412. The article states that the data opens for at discussion of the different ways professionals and users understand and solve user’s problems. That’s true. Maybe it also opens for a discussion of professional and users experience and perception of ‘coherency’. Is there a difference? It seems (from your data) that a coherent system/coherent services from a professional perspective and coherency for users are not quite the same. For users, coherency seem to be related to not only continuity between sectors and easy handovers. Coherency also seems to involve the integration of the psychiatric services/help in user’s daily life (where most of people’s recovery unfold). So again – do the authors have a preconceived idea of ‘coherent/coherency’? Does the study (also) try to explore the very experience and perception of coherency? This would be interesting and relevant. Please specify.

Interesting discussion suggestions that we have tried to unfold more.

Line 454 - Methodological considerations – is mainly about implication for practice and research – and does not consider methods much.

We have added something about the methodological. 

Line 490 – “paradigmatic approaches (biomedicine and holism…”. This is a rather simplified description – a reference is needed, at least.

We have add a reference. Line 530

I do not feel qualified to judge about the English language, but there are sentences that seems to need some kind of editing, because it is simple hard to know what the authors actually mean, e.g. line 90: “ The time in the study was established in a research group …” or line 152: “The units of opinion can be part of…” Look the text carefully through, please.

We have checked the language and sent it to English professional proofreader.

The conclusion hardly sums up users experience and perception of coherency. Please do.

It was a mistake, which we have now corrected. 

Thank you for all your constructive comments

Best regards 

Authors 

Reviewer 3 Report

This is a useful study which is spoilt by poor English expression. One suggestion is to use the Grammarly software to help make suitable changes.

Lines 124, 156 417 require reference numbers

Table 2 use indent for any theme more than 1 line long

Table 3 is confusing, use lines or some method to clearly delineate

Line 184 reveal the approval number

Table 4 is also confusing

The authors need to consider how to make the text flow in a way that facilitates ease of presentation.

Perhaps consider some diagrams, etc as a way of communicating the themes and sub-themes.

It would be nice to se a series of recommended actions to improve the situation.

Author Response

This is a useful study which is spoilt by poor English expression. One suggestion is to use the Grammarly software to help make suitable changes.

Dear reviewer

Thank you very much for constructive comment. 

- We have now tried to correct and subsequently sent it to prof english editing

Lines 124, 156 417 require reference numbers

- Is now corrected.

Table 2 use indent for any theme more than 1 line long

- Is now corrected.

Table 3 is confusing, use lines or some method to clearly delineate

- We have separated the text more so that it appears more clearly

Line 184 reveal the approval number

- Anonymized

Table 4 is also confusing

- We have separated the text more so that it appears more clearly

The authors need to consider how to make the text flow in a way that facilitates ease of presentation.

Perhaps consider some diagrams, etc as a way of communicating the themes and sub-themes.

It would be nice to se a series of recommended actions to improve the situation.

- We have given the article one more turn to strengthen the flow and dissemination